# Study on the Densification of Osmium by Experiment and First Principle Calculations

**DOI:** 10.3390/ma15228011

**Published:** 2022-11-13

**Authors:** Yunfei Yang, Junhao Sun, Wei Liu, Peng Hu, Ruimin Zhang, Hexiong Liu, Junyan Gao, Jinshu Wang

**Affiliations:** Key Laboratory of Advanced Functional Materials, Education Ministry of China, Faculty of Materials and Manufacture, Beijing University of Technology, Beijing 100124, China

**Keywords:** osmium, densification, master sintering curve, first-principle calculation

## Abstract

The sintering of osmium is critical for the preparation of raw material targets for film coating, which is the main application area of osmium. In order to get a better understanding of the intrinsic mechanism of densification of osmium, a serial study on the sintering behavior of osmium has been made in this study. By the master sintering curve (MSC) and constant heating rate (CHR) method, the sintering activation energy of nanosized osmium is evaluated to be about 340 kJ/mol, which is higher than most other metals. The density–functional theory calculation indicates the higher energy barrier of the surface atom and vacancy migration and lacking migration tunnel of inner point vacancies. For example, the diffusion of osmium atoms on the surface of particles is mainly limited by Os (1010), which has an energy barrier as high as 1.14 eV, that is higher than the W atom on W (110) of 0.99 eV. The vacancy migration energy barrier inside osmium’s grains is higher than 3.0 eV, while that of W is only 1.7 eV. This means that it is more difficult for osmium to achieve a high density compared with W, which is consistent with the experimental results. Accordingly, the proposed strategy provides a new opportunity to design a sintering process for target fabrication with excellent properties for various applications.

## 1. Introduction

Osmium and its alloys have been widely applied in electrical contacts, cathodes, fountain-pen nib tipping and other applications because of their exceptional physicochemical properties, such as chemical stability, high hardness, density, melting points, and bulk modulus [1,2,3]. In addition, osmium has also played a significant role in developing ammonia synthesis and M-N2 model chemistry. Considering its high prices, osmium coating has come to be the primary application strategy to cut down its dosage [4,5,6,7,8,9,10], and sputtering deposition technology is the most adopted for coating fabrication due to its advantage of film purity [8,9,10,11,12,13]. In particular, the sputtering target has a decisive effect on the properties of deposited thin films during the deposition process, and powder metallurgy is the best option to fabricate the osmium target due to its high melting point [14,15,16]. Nevertheless, as the crucial stage in manufacturing, the sintering of osmium has not been studied widely until now. Especially, the sintering activation energy and densification mechanism of osmium have not been evaluated, which are very important for supporting the design of the sintering process. Hence, it is imperative to investigate the densification mechanism of osmium during the sintering process.

Considering the high price of osmium, comprehensive understanding of the intrinsic sintering mechanism is impractical to obtain exclusively through experimental work. Alternatively, predicting the densification behavior by theoretical calculation is a feasible way to clarify the relationship between the sintering process and the structure of the obtained alloy, and many theoretical explorations, such as numerical analysis and finite element analysis, have been conducted to investigate the densification mechanism of powder metallurgy [17,18,19]. Among these works, the master sintering curve analysis has revealed its advance to efficiently assess activation energy and characterize the sintering behavior of different materials, which is mainly based on diffusion theories and experimental results of the sintering parameters to the relative density at any point in the sintering procedure (regardless of sintering time and temperature). Up until now, this method has been applied to the sintering investigation of many materials, such as molybdenum, tungsten, nickel, tungsten–molybdenum–copper alloy [20,21], as well as ceramics and glasses such as alumina, zirconia and so on [22,23,24,25,26]. Thus, investigation of the densification mechanism of osmium powder by master sintering curve is highly desired to fulfil the ever-growing requirement in various fields.

In this study, the densification behavior of osmium nanopowder is theoretically investigated based on experimental sintering results. With the method of the master sintering curve (MSC) and constant heating rate (CHR) model, the sintering activation energy of osmium is evaluated. Furthermore, the migration energy barriers in multiple paths of osmium are comparatively studied by first principle calculation. The obtained results facilitate a better understanding of the characteristic features of the sintering densification process of osmium nanopowder, and also provide an opportunity for predicting the sintering behavior of other materials.

## 2. Experiment and Calculation

The experimental raw material is pure osmium powder (99.95% purity and average particle size of 20 nm) produced by Beijing Jiaming Platinum Nonferrous Metals Co., Ltd. In order to reduce the influence of residual oxygen on sintering, the powder was firstly reduced in hydrogen at a temperature of 650 °C and then ground. The purified powders were consolidated to a cylindrical green pellet (3 mm in diameter and 1.5 mm in thickness) using a hydraulic press with pressure of 700 Mpa. The green density of the pressed sample was calculated by measuring the geometrical dimension of the samples using a Vernier caliper. In the sintering process, the green bodies were heated in a tungsten rod furnace at different heating rates of 1 °C/min, 5 °C/min, 10 °C/min, 20 °C/min, 30 °C/min, 40 °C/min from room temperature to target temperature, respectively, and dry hydrogen was used as protective atmosphere. Meanwhile, the cooling rate was set as 200 °C/min from terminal temperature to 500 °C to avoid possible influence of relative density during the cooling process.

Density functional theory (DFT) calculations were carried out with Vienna Ab initio Simulation Package (VASP) [27,28]. The generalized gradient approximation (GGA) was used to describe the exchange–correlation interactions, and the Perdew–Burke–Ernzerhof (PBE) functional was used as an exchange–correlation functional approximation [29,30,31]. The plane wave cut-off energy was fixed at 500 eV, and the convergence threshold was 10^−7^ eV. The atoms were relaxed until the force acting on each atom was less than 0.01 eV/Å. The vacuum layer thickness was set to be more than 20 Å for subsequent calculations, which was sufficient to eliminate the influence of adjacent systems. Only the gamma k-points grid was used to describe the Brillouin zone for geometric optimization and self-consistent calculations [32]. The energy barriers were calculated by the climbing image nudged elastic band (CI-NEB) method [33,34,35,36]. For CI-NEB calculation, cut-off energy is set to 450 eV, and the break condition for the ionic relaxation loop is −0.1 eV/Å.

## 3. Results

### 3.1. Morphological Characterization

Appendix A show the XRD and TEM images of purified powders, respectively. They indicate osmium is a HCP structure and has not been oxidized. Although agglomerated osmium powder was dominant, the particles kept a small size of about 20 nm after purification. The fracture surface images of the samples sintered at different temperatures are illustrated in Figure 1. It can be seen that the grain sizes of osmium were gradually coarsened during the densification process. In Figure 1ax–ex, the samples with terminal temperature of 1300 °C possess average grain sizes of 240 nm–500 nm with the decreasing heating rate of 40 °C/min to 1 °C/min, which is dozens of times larger than the initial particle size of raw osmium powder. The obvious grain growth should be attributed to the strong activity of nanosized osmium powder. As shown in Figure 1ax–ez, the grain continually grew at 1500 °C, 1700 °C, with further increased sintering temperature at different heating rates. It is also noticed that the grain size is positively related to the density of samples even at a different heating rate, for instance, samples in Figure 1by,ez. It means that the grain growth of osmium follows a single path, which only depends on density instead of other parameters such as thermal sources or cycles.

In addition, the average pore size firstly increased from 20 nm to 40 nm when the density increased from 70.42% to 76.43%, but the number of pores decreased, as shown in Figure 1dx,ex. Further increasing the sintering temperature, both the pore size and the number decreased, as shown in Figure 1c,d, indicating that the pores have experienced a process of first merging and then contracting during sintering. This evolution of the pore number is consistent with the densification process, in which continuous shrinking was observed in the obtained sample.

### 3.2. MSC of Nanosized Osmium

#### 3.2.1. Theoretical Background: Master Sintering Curve Analysis

To further understand the densification behavior of osmium nanocrystals, the sintering apparent activation energy (*Q*) is calculated by MSC fitting [37,38,39,40,41,42]. The way to obtain the apparent activation energy *Q*_MSC_ in the MSC is to make an estimate value *Q* and then construct the curves of *ρ* versus log(*Θ*) with the different heating processes. Herein, the relative density (*ρ*) is the ratio of the sintered density *ρ_s_* to the theoretical density *ρ_th_*.
(1)$$\rho =\frac{\rho_{s}}{\rho_{th}}$$

The *Θ* is sintering work that is derived from Equation (2).
(2)$$\Theta \left(t,T\right)=\int_{t_{0}}^{t}\frac{1}{T}\exp\left(-\frac{Q}{kT}\right)dt$$

The appropriate *Q* makes all the curves with different heating processes converge to a single curve, according to the MSC theory. A Boltzmann sigmoid curve is set to describe the single curve.
(3)$$\rho =\rho_{0}+\frac{1-\rho_{0}}{1+\exp\left(-\frac{log\left(\Theta \right)-b}{c}\right)}$$

#### 3.2.2. MSC Results of Nanosized Osmium

Figure 2a displays the relative density of the samples as the function of sintering temperature and time at different heating rates. The decrease in slopes, as there is an increase in heating rates, demonstrates the inverse relationship between densification speed and heating rate, typically, for most sintering cases. Figure 2b shows the curves of relative density vs. holding time at various temperatures. It can be seen that increased density of the sample could be achieved by prolonging the holding time. The relative density of the sample is identified to be about 94% after being sintered at 1500 °C for 20 min and kept almost stable by further increasing the holding time. The samples sintered at 1450 °C and 1400 °C exhibited the same sintering behavior, since the diffusion path, such as surface and interface which has low energy barriers, decreased gradually with the pore closure and grain growth. Since the temperature is much lower than the melting point, atoms and vacancies find it hard to overcome the high energy barriers of volume diffusion, resulting in an extremely slow rate of diffusion and densification in the later stage during the isothermal sintering.

Figure 3a displays the fitted curve on *ρ* vs. log(*Θ*). As shown in Figure 3a, the data obtained at different heating rates fit well with the curves at *Q*~300 and *Q*~400, indicating that the actual apparent activation energy of the samples is within 300–400 kJ/mol (3.125–4.17 eV/atom), which is well in accordance with the values of sintering apparent activation energy (in the range of 150–550 kJ/mol (1.56–5.73 eV/atom)) for refractory metals, according to previous studies [43,44,45,46].

Figure 3b shows the relationship between the mean residual square (MRS) and apparent activation energy *Q*_MSC_, where MRS is calculated by Equation (4).
(4)$$MRS=\sqrt{\frac{1}{\rho_{f}-\rho_{0}}\int_{\rho_{0}}^{\rho_{f}}\frac{\sum_{i=1}^{N}{\left(\frac{\Theta_{p,i}}{\Theta_{p}}-1\right)}^{2}}{N}d\rho }$$

According to the fitted curve in Figure 3b, the smallest mean residual square appeared at *Q_MSC_* = 340 kJ/mol (3.54 eV/atom), which presents the real activation energy of osmium. From previous studies, tungsten powder with particle size from 100 nm to 1 um generally has sintering apparent activation energy of 250–350 kJ/mol (2.60–3.46 eV/atom) [45,46,47], which is closed to our result of osmium powder with a size of 20 nm, indicating a higher sintering activation energy of osmium than that of W with the same particle size.

Based on the calculated *Q_MSC_*, the prediction map of samples for various heating rates was built and is shown in Figure 4. As indicated in the 3D map, the densification process came to be more slowly at the same termination temperature with increased heating rate. Meanwhile, it can be seen that the map can be divided into three sections according to the temperature zone: the slowly increased density at low temperature (Area I), then the rapidly grown density with increased temperature (Area II), and densification significantly slowing down in the third section.

The expected final density of the sintered samples at different conditions was calculated using the MSC, which is shown in Table 1. It is found that the predicted density fits well with the experimental values in the middle and last stages but deviated in the initial stage. For examples, the sample sintered at 1000 °C with a heating rate of 1 °C/min has the relative density of 51.43%, while the predicted density by MSC is only 48.65%, which is much lower than the experimental results. However, the predicted values of samples sintered at 1500 °C have errors less than 1%. That is because only the volume diffusion and grain boundary diffusion are taken into consideration in the MSC model [19], while surface diffusion, as the main mechanism at low temperature, has not been fully considered. Therefore, considering the instantaneous change of activation energy with the grain growth and densification during sintering, the MSC model cannot fully meet the precise prediction of the full temperature range.

#### 3.2.3. Densification Mechanism of Osmium

Except for MSC, the constant heating rate (CHR) model is used to evaluate the apparent activation energy (*Q_CHR_*) during the sintering process, which could be given as the following equation:(5)ln[TC(dρdT)]=−QRT+ln(f(ρ))+lnA−nlnG
where *G* is the grain size, *n* is the gain size exponent depend on the dominant diffusion mechanism, *A* is a material parameter independent of temperature *T*, *R* is the gas constant, *f(ρ)* is a function of density. At different heating rates, *C*, with a measured value of *ρ*, the left side of Equation (5), against the reciprocal temperature 1/*T*, would be a straight line with the slope which estimates the value of *Q_CHR_*.

The relationships between *ln[TC(dρ/dT)]* and *1/T* for all samples at five different heating rates for both microwave and conventional sintering methods are demonstrated in Figure 5. It is clear that the straight lines obtained are approximate but incompletely parallel. According to the slope of the straight lines, the different apparent values for *Q_CHR_* increased from 321.95 kJ/mol (3.35 eV/atom) to 335.40 kJ/mol (3.49 eV/atom) with the increasing relative density from 50% to 60%. The increase of activation energy in the initial stage of densification can be attributed to the growth of grains. While, with the further increasing of density, the *Q_CHR_* came to be about 340 kJ/mol (3.54 eV/atom) with relative density of 70–90%, which is close to the *Q_MSC_* of osmium.

To further study the diffusion mechanism, Equations (6) and (7) as follows are applied to determine the atom migration during sintering.
(6)$$\frac{\Delta L}{L_{0}}=A\left(T\right)t^{1/n}$$
(7)$$\ln\frac{\Delta L}{L_{0}}=lnA\left(T\right)+\frac{1}{n}lnt$$
where Δ*L/L_0_* is the relative shrinkage, *A(T)* is a constant related to the sintering temperature, *t* is the sintering time, and *n* is the sintering characteristic index. The atom migration mechanism in the sintering process can be determined according to the value of *n*.

Relative shrinkages and sintering kinetic curves of osmium samples at different temperature are shown in Figure 6. By calculating the slope of the curve in Figure 6b, the sintering characteristic index n is obtained and shown in Table 2. The value n of osmium should be around 3.49–4.48 at 1500 °C, and is obviously larger than 3.0. This means that boundary diffusion dominates the atom migration. Meanwhile, it can be found that the value of n slightly proportionally increases with the relative density, as n_1_, that is calculated with shrinkage in a shorter insulation process time, is lower than n_2_, with longer insulation time and higher density. Additionally, with the increasing of temperature from 1400 °C to 1500 °C, the value of n increased from 3.49 to 4.48 with the increase in the relative density. Considering the high melting point and density of osmium, volume diffusion is difficult to activate at lower sintering temperatures, so that grain boundary diffusion dominates the main way of atom migration.

However, it is an indisputable fact that the sintering activation energy is affected by the differences in models, calculation methods, and source material, and hard to direct and provide an accurate comparison. In order to determine the sintering difficulty of osmium more objectively, the thermodynamic and kinetic analysis related to osmium sintering was carried out using first principles.

### 3.3. First Principle Study on Densification of Osmium Nanocrystals

#### 3.3.1. Thermodynamic Analysis of Densification Depending on Surface Energy

The apparent activation energy is generally adopted to describe the sintering behavior of the sample at macro level. Actually, mass migration plays an important role in the densification process. So, it is necessary to investigate the densification mechanism of osmium nanocrystals from the atom point of view, and the energy barriers related to the sintering of osmium are investigated in this part and compared with that of W, since they have similar atomic size and melting point.

The driving force of sintering appeared as differences in bulk pressure, vacancy concentration and vapor pressure due to the differences in surface curvature of the particles. According to the thermodynamic concept, the sintering pressure is described as the change in the total free energy with respect to total volume change during densification, which can be calculated as [47]:(8)$$P_{0}=\frac{2\gamma_{b}}{G}+\frac{2\gamma_{s}}{r}$$
where γ_b_ and γ_s_ are the grain boundary and surface energy, respectively, *G* is the grain size, and r is the radius of neck curvature. Because *γ_s_ > γ_b_* and *G > r* for a common situation, Equation (9) can be diverted as:(9)$$\Delta P=\frac{2\gamma_{s}}{r}$$

So, the sintering driving force can be reduced to the difference in surface energy.

The surface energy of the typically exposed crystal plane of osmium and W is calculated with Equation (10), and the results are shown in Figure 7a.
(10)$$\gamma_{s}=\frac{E_{slab}-nE_{bulk}}{2A}$$
where *E_slab_* is the total energy of a slab system, n is the number of atoms of a slab model, *E_bulk_* is the chemical potential of an atom in the bulk phase, and *A* is the surface area [48].

It is noticed that the (0001) plane of osmium crystal, as with most close-packed planes, has the lowest surface energy of 0.19 eV/Å^2^; and surface energy of different crystal planes of osmium are in the order of (1120) > (1010) > (0001), which are similar to those of Ref. [49]; while for W, the close-packed planes of (110) have a surface energy of 0.20 eV/ Å^2^. However, it is difficult to judge the difference between the surface energies of W and osmium from the above results, since the proportion of exposed crystal planes of osmium on the powder surface could not be well defined. In order to accurately describe the surface energy of osmium and W, the Wulff construction, as shown in Figure 7b, is constructed based on the above calculation results of the surface energy [50,51,52].

According to statistical results, Os (0001) takes the 26.19% surface area, less than 42.29% of (1011) and 30.52% of (1010), and weighted mean surface energy should be 0.22 eV/A^2^ from the calculated results (shown in Figure 7a). While for W, the close-packed plane of (110) has a dominant occupation of exposed surface energy with more than 60%, and the rest mostly are (112) and few (100). The surface energy per area of W is 0.21 eV/A^2^, close to that of osmium, indicating the similar driving force of sintering of osmium and W when they have the same specific surface area.

#### 3.3.2. Kinetic Analysis of Densification by Vacancy Migration Model through Multiple Paths

On the other hand, sintering densification is highly dependent on the kinetic conditions of solid-phase atom migration, since the sintering temperature of refractory metals is much lower than their melting points. In the initial stage of sintering, surface diffusion is the main route of atom migration, in which the sintering activation energy is mainly determined by the energy barrier. In order to evaluate the surface diffusion rate, the energy barrier of self-surface diffusion is calculated and shown in Figure 8.

According to the calculated results, the migration of the W atom on W (110) surface has an energy barrier of 0.99 eV, which is higher than 0.90 eV on (0001) plane and 0.65 eV on (1011) plane of osmium, respectively, but lower than 1.14 eV on (1010) plane of osmium. Wulff construction, shown in Figure 8b, Os (1010) takes nearly 30.52% of the exposed surface and thus serves as the essential channel for mass migration through the surface. Hence, although (0001) and (1011) planes of osmium have a lower diffusion barrier than W (110), the surface diffusion of osmium is limited due to the high diffusion energy barrier of the exposed (1010) plane.

For the last stage of sintering, interface diffusion and volume diffusion play an important role in the proceeding of densification. At this time, pores would move through vacancies in grains and grain boundaries to accomplish the densification. Since there is a high complexity of the interface in sintered samples, it is difficult to obtain representative interface states through the characterization of the micro-region for calculating energy barriers of vacancy diffusion. However, in the process of vacancy diffusion, the energy barrier is mainly determined by the interaction of atoms around the vacancy, while at the dense region with stronger bond, the diffusion becomes more difficult. Therefore, the diffusion energy barrier at the interface should be higher than that in the top layer and smaller than inner crystals due to its insufficient compactness [53,54,55]. Herein, the energy barriers for vacancies migration in the top layer of the slab and inside the crystal are calculated to evaluate the difficulty of interface and volume diffusion.

Figure 9 displays the surface vacancy diffusion model of Os (0001), Os (1011), Os (1010) and W (110). The energy barriers of vacancy diffusion in the top layer of Os (0001)/(1010)/(1011) are 2.41 eV, 2.50 eV and 1.85 eV, respectively, while at W (110) is 1.41 eV. The large gap of energy barriers for osmium and W indicates that the diffusion of surface vacancy of osmium is suppressed and harder to obtain high densification compared to W. This also implies that the vacancy migration energy of osmium at the interface may also be higher than that of W. Because the vacancy migration at the interface is simultaneously affected by the two surfaces, while the surface vacancy diffusion energy barrier of osmium is generally higher than W, this will greatly hinder the degassing process during sintering densification.

Investigation of the diffusion barrier of vacancy inside the lattice was further carried out, and the results are shown in Figure 10. Compared with an atom moving in a specific plane, the vacancy migration in the crystal is involved in many migration paths. For HCP structure, there are two coordination of the vacancy center and surrounding atoms, which correspond to two diffusion paths. As indicated in Figure 10a, the first route is the vacancy diffusing along the <0001> directions to accomplish diffusion in {0001} planes, which has an energy barrier of 3.08 eV. Another is the [2203] direction which can implement atom transitions between {0001} layers that have an energy barrier of 3.21 eV. Herein, HCP {0001} planes have to be mentioned because the {0001} plane family is parallel to each other in the whole crystal; the second route [2203] is obviously an indispensable route for the random movement of vacancies in the grain. Just as shown in Figure 10d, it is a feasible way of vacancy movement in osmium that involves diffusion along <0001> and <2203>.

As shown in Figure 10b, the vacancy is compassed with two perpendicular {110} planes for BCC tungsten. The positions of surrounding atoms of the vacancy are mainly summarized as point A ({001} direction) and B ({111} direction), and the corresponding distances to the vacancy center are about 2.76 A and 3.19 A, respectively. When vacancy diffuses along <111> directions, the energy barrier, as shown in Figure 10c, is 1.70 eV, which is not only much lower than that along with <001> directions (5.95 eV), but also much lower than the two diffusion routes of osmium. Meanwhile, it is noticed that the {110} family has eight equivalent planes which are parallel or perpendicular to each other due to the high symmetry of BCC structure. Among them, perpendicular (110) and (011) could give multiple routes, along with <111>, to obtain the three-dimensional diffusion. Hence, differently from osmium, inside vacancies have nearly 3 eV energy barriers for diffusion along multiples paths; the vacancies in W could migrate through “tunnels” formed by staggered <111> orientation that has an extra low energy barrier to accomplish the densification, as shown in Figure 10e. Meanwhile, the inner defect diffusion energy barriers (nearly 3.08 eV) are close to the MSC and CHR calculated results of 340 kJ/mol (3.54 eV/atom).

In summary of the calculation results shown in Table 3, osmium and W particles commonly have similar sintering driving forces, since these two metal grains have closed surface energy in thermodynamic equilibrium states. However, due to the higher energy barrier of the surface atom migration and surface vacancy migration, and lack of migration tunnel of inner point vacancies, the densification of osmium should be much harder than that of W. While according to the previous research on the densification of W, the sintering activation energy of nanosized W is commonly in the range of 100–300 kJ/mol (1.042–3.125 eV/atom), which is smaller than the calculation results of osmium in our studies. It is suggested that the calculations of vacancy migration are consistent with the experimental results, qualitatively.

## 4. Conclusions

The sintering behaviors and densification mechanism of osmium nanopowder were experimentally and theoretically investigated in this work. By constructing the master sinter curve, along with a constant heating rate method, the sintering activation energy of osmium nanopowder is identified to be 340 kJ/mol (3.54 eV/atom), larger than that of tungsten powder with the same particle size. In addition, a first principle calculation was carried out to identify the intrinsic mechanism of the densification process. It is found that the migration of osmium atoms on the surface is mainly limited by the inevitably exposed Os (1010) plane with an energy barrier of 1.14 eV, that is not only higher than Os (0001) and Os (1011), but also higher than that of W atoms migrating on W (110). Furthermore, the migration energy barrier of vacancy in the surface layer and inner grains of osmium is also higher than that of W. Especially within the grain, vacancy migration in osmium is higher than 3.0 eV for both <0001> and <2203> directions. While in tungsten, vacancy migrating with <111> only has an overall energy barrier less than 1.8 eV. The results provide strong guidance for designing the sintering and densification process of osmium, and also have theoretical significance for the sintering and densification of other HCP metals.

## Figures and Tables

**Figure 1 materials-15-08011-f001:**
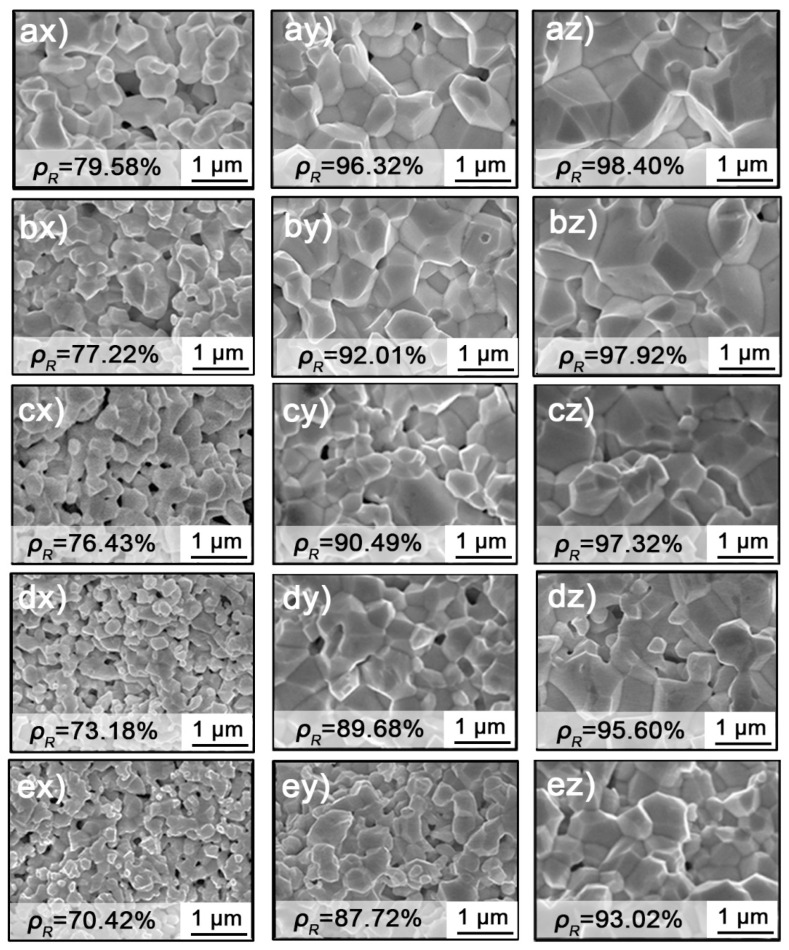
Samples sintered at different temperatures (**x**–**z**) 1300, 1500, 1700 °C (hold for 10 min) and multiple heating rate (**a**–**e**) 1, 5, 10, 20, 40 °C/min with different relative density.

**Figure 2 materials-15-08011-f002:**
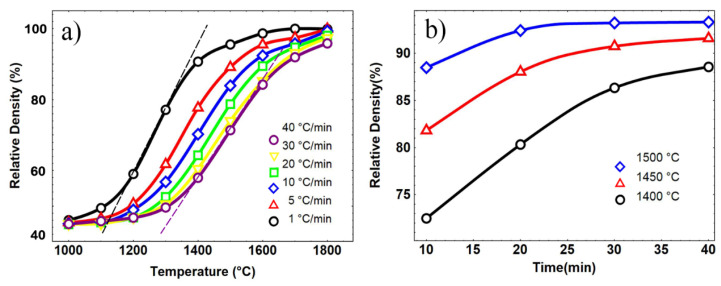
Densification tendency of samples prepared at (**a**) different heat-ramp rate, (**b**) isothermal duration at different temperatures.

**Figure 3 materials-15-08011-f003:**
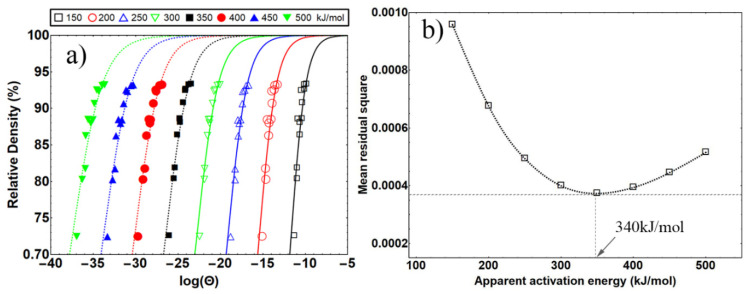
(**a**) Construction of the *ρ*-log(*Θ*) curves with different apparent densification activation energy. (**b**) Mean residual squares of the master sintering curve for different *Q* values.

**Figure 4 materials-15-08011-f004:**
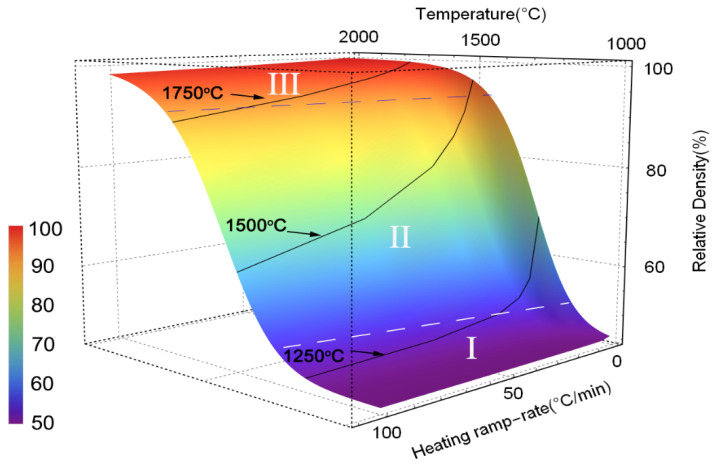
Predicted densification map using master sintering curve.

**Figure 5 materials-15-08011-f005:**
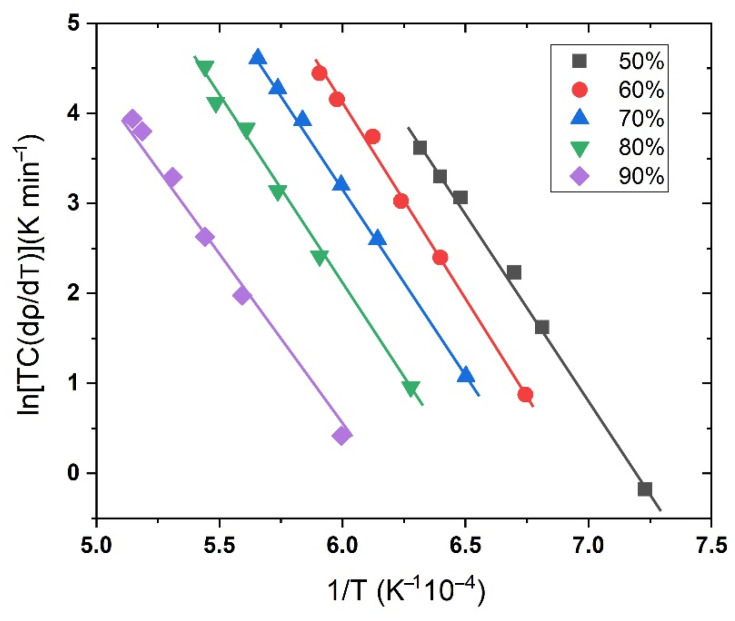
The Arrhenius plot of *ln[TC(dρ/dT)]* vs. *(1/T)* for the non-isothermal sintering of the powder compacts at different heating rates corresponding to the same relative density.

**Figure 6 materials-15-08011-f006:**
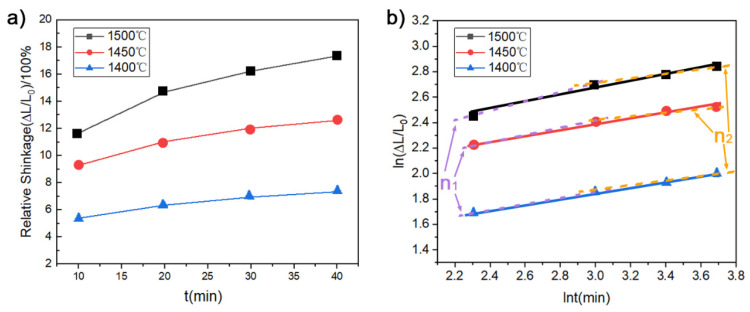
Os samples sintered at different temperatures (**a**) Relative shrinkages; (**b**) sintering kinetic curves.

**Figure 7 materials-15-08011-f007:**
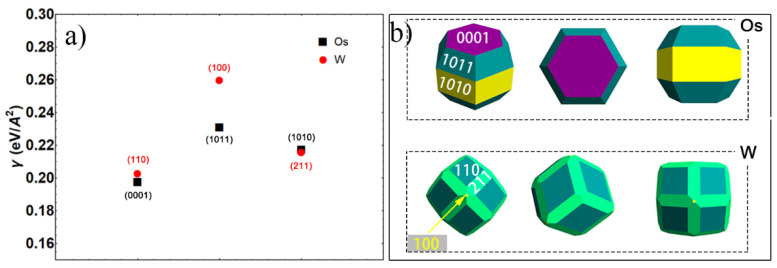
Comparison of (**a**) surface energy and (**b**) equilibrium Wulff shape of Os and W.

**Figure 8 materials-15-08011-f008:**
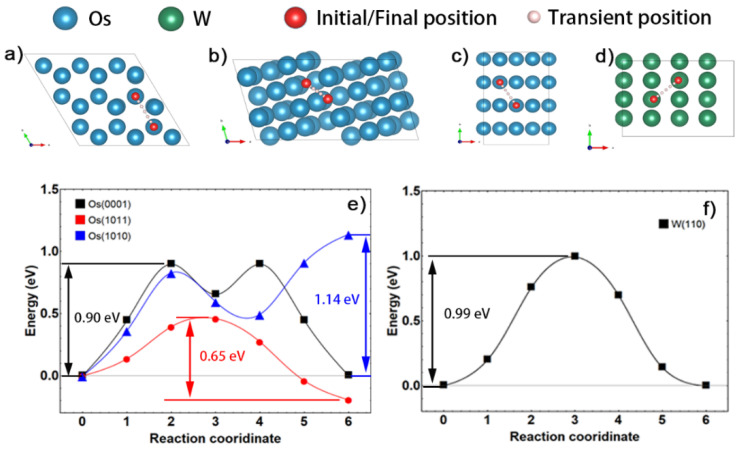
Models of surface atom migration of (**a**–**c**) Os on Os(0001)/(1010)/(1011) and (**d**) W on W(110), energy barriers of (**e**) Os on Os surface, (**f**) W on W(110).

**Figure 9 materials-15-08011-f009:**
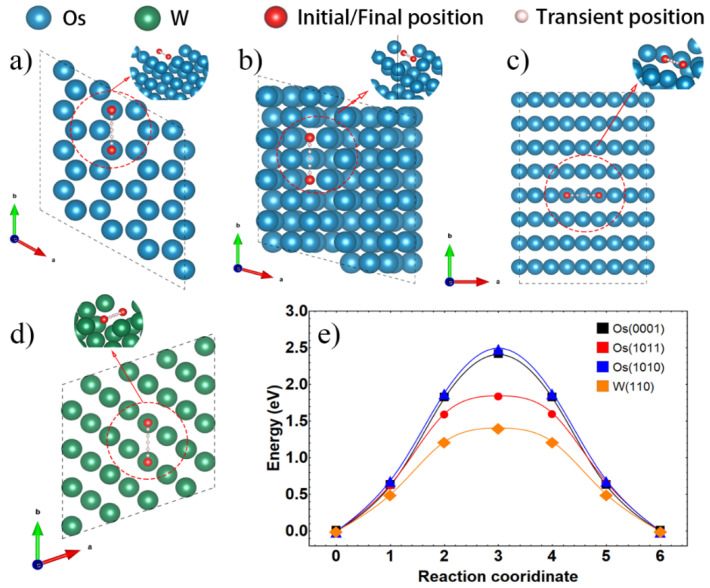
Models of surface vacancy migration of (**a**–**c**) Os on Os(0001)/(1010)/(1011) and (**d**) W on W(110), (**e**) the corresponding energy barriers.

**Figure 10 materials-15-08011-f010:**
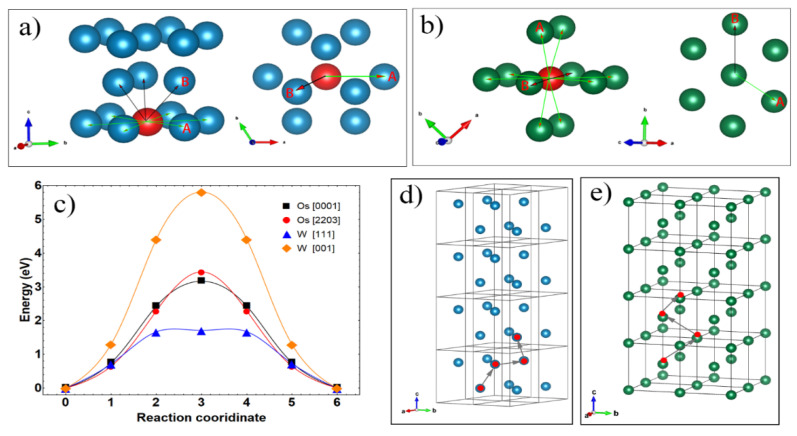
Inner vacancy migration of (**a**) Os with direction A <0001> and B<2203>, (**b**) W with direction A <001> and B<111>, (**c**) energy barriers, (**d**) path example of Os, and (**e**) path example of W.

**Table 1 materials-15-08011-t001:** The experimentally measured and predicted relative density using master sintering curves for different sintering procedure.

Heating Rate °C/min	Termination Temperature °C	Relative Density (%)	Predicted Density (%)
1	1100	51.43	48.65
1	1500	95.55	96.09
1	1700	97.97	98.31
5	1100	50.49	46.31
5	1500	89.12	89.29
5	1700	96.45	97.38
10	1100	49.38	45.80
10	1500	83.93	84.37
10	1700	95.80	96.60
20	1100	48.38	45.49
20	1500	78.73	78.29
20	1700	94.73	94.63
40	1100	47.75	45.30
40	1500	71.38	71.55
40	1700	91.65	91.70

**Table 2 materials-15-08011-t002:** Values of n in the sintering kinetic equation.

T/°C	1400	1450	1500
n	3.49	4.24	4.48
n_1_	2.94	3.92	4.08
n_2_	4.31	4.76	4.83

**Table 3 materials-15-08011-t003:** Energy barrier of atom and vacancy migration on surface and in lattice of Os and W.

	Surface Atom		Surface Vacancy		Inside Vacancy	
	Plane	Energy Barrier (eV)	Plane	Energy Barrier (eV)	Direction	Energy Barrier (eV)
W	(110)	0.99	(110)	1.41	<001>	5.95
					<111>	1.70
Os	(0001)	0.90	(0001)	2.41	<0001>	3.08
	(1010)	1.14	(1010)	2.50	<2203>	3.21
	(1011)	0.65	(1011)	1.85	/	/

## Data Availability

Not applicable.

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
