# Peer review of "Study on the Densification of Osmium by Experiment and First Principle Calculations"

_materials, 2022, doi:10.3390/ma15228011_

Round 1

Reviewer 1 Report

This manuscript is clear, easy to read, and presents very interesting results.Also, the introduction is clear and to the point. However, there are a few unclear issues related to the use of MSC; thus, I believe a MAJOR REVISION is required. 

The main issue: 

although constructing master sintering curves is a practical and powerful approach to comparing and predicting the sintering behaviour of materials, it is not the best method to determine the sintering mechanism and estimate the corresponding activation energy. Therefore, it is very unlikely that the combination of MSC and First Principle Calculations results in a conclusive outcome. 

Often it is suggested to exploit the MSC to predict sintering behaviour which is more related to what is mentioned in the second paragraph of the introduction of this manuscript. Alternatively, more detailed methods ( eg CHR) should be used to measure the sintering activation energy and its variations over the sintering window. 

I believe using one of the above-mentioned lines makes this manuscript a significant contribution in the field of sinteirng. 

Moreover, there are afew minor issues: 

Use consistent units of energy: 

eV or kJ.mol, and if it is not possible ( in case of comparison, give the number) 

line 66: it is better to report the pressure in MPa ( using SI units) 

section 3.2)

(a) it is better not to call the values measured values by constructing MSCs activation energy of sintering since Q is an oversimplified value. ( probably, apparent activation energy is a bit better) 

(b) The construction of MSC is based on the assumption that the microstructure is not influenced by thermal history; on the basis of this assumption, one can obtain Eq 2. Thus, it is suggested to mention the role of microstructure. Also, it is better to show the microstructure of samples after sintering ( supporting the assumption) 

The purpose of Fig 4 is not clear to me. In fact, plotting relative density as a function of log theta shows the sintering behaviour of the material. Moreover, the deviations of the experimental results from the constructed curve point out the changes in the sintering mechanism ( as the authors highlight this point in the end of the paragraph). Please see https://doi.org/10.1016/j.jeurceramsoc.2013.01.020  figure 4. 

Author Response

Response to Reviewer 1 Comments

Dear reviewers:

Thank you for your comments on our manuscript entitled " Densification Mechanism of Os: Master Sintering Curve Fitting and Frist Principle Calculation". Those comments are very helpful for revising and improving our paper, as well as the important guiding significance to other research. We have studied the comments carefully and made corrections which we hope meet with approval. The main corrections are in the manuscript and the responds to your comments are as follows.

Point 1: Although constructing master sintering curves is a practical and powerful approach to comparing and predicting the sintering behaviour of materials, it is not the best method to determine the sintering mechanism and estimate the corresponding activation energy. Therefore, it is very unlikely that the combination of MSC and First Principle Calculations results in a conclusive outcome. 

Often it is suggested to exploit the MSC to predict sintering behaviour which is more related to what is mentioned in the second paragraph of the introduction of this manuscript. Alternatively, more detailed methods ( eg CHR) should be used to measure the sintering activation energy and its variations over the sintering window. 

I believe using one of the above-mentioned lines makes this manuscript a significant contribution in the field of sinteirng. 

Response:According to your recommendations, we have analyzed the apparent activation energy with method of Constant Heating Rate(CHR) in section ‘3.2.3 Densification mechanism of Osmium’. Meanwhile, the diffusion mechanism also been studied and shown in this section. At the same time, for these changes in the manuscript, the title has been changed to ‘Study on the densification of Osmium By Experiment and First Principle Calculations’

Point 2. Use consistent units of energy: eV or kJ.mol, and if it is not possible ( in case of comparison, give the number) 

line 66: it is better to report the pressure in MPa ( using SI units) 

Response:

Thank you for your rigorous suggestion, because the unit of activation energy in previous experimental studies is mainly kJ/mol, while the first-principles calculation is mainly in eV/atom. After consideration, we marked the value corresponding to eV/atom after the kJ/mol value in manuscript.

Point 3. It is better not to call the values measured values by constructing MSCs activation energy of sintering since Q is an oversimplified value. ( probably, apparent activation energy is a bit better) 

Response:

Thank you for your rigorous suggestion. We have modify MSCs activation energy to be the apparent activation energy QMSC, yet CHR activation energy as QCHR to distinguish them.

Point 4. The construction of MSC is based on the assumption that the microstructure is not influenced by thermal history; on the basis of this assumption, one can obtain Eq 2. Thus, it is suggested to mention the role of microstructure. Also, it is better to show the microstructure of samples after sintering ( supporting the assumption) 

Response:

As stated in this manuscript, predicts accuracy of MSC in low-density stage are obvious lower than that of high-density stage. The reason should be the inevitably microstructural variation during Initial stage of sintering. Therefore, in response to your suggestion, we have modified Figure 1 and added more characterization results for sintered samples.

Point 5. The purpose of Fig 4 is not clear to me. In fact, plotting relative density as a function of log theta shows the sintering behaviour of the material. Moreover, the deviations of the experimental results from the constructed curve point out the changes in the sintering mechanism ( as the authors highlight this point in the end of the paragraph). Please see https://doi.org/10.1016/j.jeurceramsoc.2013.01.020  figure 4.

Response:

Sorry for the inconvenience of understanding. As you know, our original intention in this study includes giving a prediction of sintered density of Os by MSC method. Hence, Figure 4 shows the corresponding relationship between the density of the samples and different heating rates and temperatures when continuous heated based on the calculated activation energy. Reference [21] in the manuscript also show a similar figure in it. And the diagram you mentioned plotting relative density as a function of log theta is shown in the Fig.3 in this manuscript.

Once again, thank you very much for your comments and suggestions.

Reviewer 2 Report

It is a scientific manuscript about the densification mechanism of osmium nanopowder. This article includes first principles calculations and densification mechanism.

Nevertheless, the authors send a version that can be considered as a first version and so many changes are needed. Thus, I suggest a major revision.

Supplementary figure: I suggest to divide this figure. It is very difficult to check XRD diffraction pattern.

Likewise, I recommend adding comments about XRD and TEM analysis of the specimens.

Please, indicate the osmium supplier company.

Section 3.2.: The sintering activation energies (300-400 kJ/mol) are compared with previous works (150-550 kJ/mol). The ranges are very different. Why? Please, improve discussion.

Section 3.3.1.: Why the surface energy per area should be 0.22 eV/A2.

There are references given in block. It is recommended to separate more. For example, in [20-26] separate and displace each type of specific material: molybdenum, tungsten, nickel, ..., alumina, zirconia.

The figure captions should be self-consistent. As an example, in the figure 1. 70.42%? 81.59%?.

Figure 6: It is difficult to read information from e) and f)

There is a big amount of typography mistakes. “Frist” in the title, “ig.1” in the 3.1. section. The authors send this manuscript without a last lecture.

The numbers of the equations should be all in the same position.

The reference format should be checked and adapted to the guidelines of this journal.

Author Response

Response to Reviewer 2 Comments

Dear reviewers:

Thank you for your comments on our manuscript entitled " Densification Mechanism of Os: Master Sintering Curve Fitting and Frist Principle Calculation." Those comments are constructive for revising and improving our paper and the essential guiding significance to other research. We have studied the comments carefully and made corrections which we hope meet with approval. The primary corrections are in the manuscript, and the responses to your comments are as follows.

Point 1. It is a scientific manuscript about the densification mechanism of osmium nanopowder. This article includes first principles calculations and densification mechanism. Nevertheless, the authors send a version that can be considered as a first version and so many changes are needed. Thus, I suggest a major revision.

Response:

We are very sorry for the inconvenience caused by our mistake. We have revised the text and other issues and hope to get your approval;

Point 2. Supplementary figure: I suggest to divide this figure. It is very difficult to check XRD diffraction pattern. Likewise, I recommend adding comments about XRD and TEM analysis of the specimens. Please, indicate the osmium supplier company.

Response:

Thank you very much for your suggestion. These mistakes have been revised according to your suggestion. Please see the new support file for details.

Point 3. Section 3.2.: The sintering activation energies (300-400 kJ/mol) are compared with previous works (150-550 kJ/mol). The ranges are very different. Why? Please, improve discussion.

Response:

As you know, there are few references for Os sintering at present. Therefore, we need to compare Os with other refractory metals and give some basic information about Os. We have checked the relevant content and sorted out the sintering activation energy of other refractory metals in the range of 150-550 kJ/mol, such as W and Mo. While as to our calculation, Os has a sintering activation energy of around 300-400 kJ/mol, which is close to other refractory metals.

Point 4. Section 3.3.1.: Why the surface energy per area should be 0.22 eV/A2.

Response:

We are sorry to have confused you, as we didn't explain it clearly. The surface energy of 0.22eV/A2 is a weighted mean surface energy that is calculated according to the exposed ratio and local surface energy of different planes. Minor changes have been made to this part of the manuscript.

Point 5. There are references given in block. It is recommended to separate more. For example, in [20-26] separate and displace each type of specific material: molybdenum, tungsten, nickel, ..., alumina, zirconia.

Response:

Thank you very much for your suggestion. These questions have been modified in the manuscript.

Point 6. The figure captions should be self-consistent. As an example, in the figure 1. 70.42%? 81.59%?.

Response:

Thank you very much for your suggestion, Since most of our samples were used to test the density and porosity, and some new content about the sintering activation energy and mechanism are added in the manuscript, we added some characterization pictures in Figure 1. But it should be noticed that this sample does not correspond exactly to the MSC process and just shows the correspondence between the morphology and density of samples from different processes.

Point 7. Figure 6: It is difficult to read information from e) and f).

Response:

Thank you very much for your suggestion. Fig.6 (e) and (f) has been enlarged in the last manuscript.

Point 8. There is a big amount of typography mistakes. “Frist” in the title, “ig.1” in the 3.1. section. The authors send this manuscript without the last lecture. The numbers of the equations should all be in the same position. The reference format should be checked and adapted to the guidelines of this journal.

Response:

We are very sorry for the detailed misidentification in the manuscript and thank you very much for your meticulous inspection. We have carefully checked the grammar and format of the manuscript and hope to gain your approval.

Reviewer 3 Report

This manuscript has studied the densification behavior or the sintering mechanism of Osmium powder particles (a metallic element) by theoretical methodology using a master sintering curve analysis. Per introduction, this is the first time the densification behavior of Osmium powders is being studied using theoretical methodology (i.e., master sintering curve analysis) instead of experimenting, which in turn provides some originality for the work. However, below is a summary list of suggestive minor revisions which may help improve the manuscript.

1.       I would suggest using the full name of the Osmium element throughout the manuscript instead of abbreviation “Os”. This include the title and abstract of the manuscript.

2.       I would suggest separating the theoretical background, i.e., 3.2. MSC of Nano-sized Osmium, from the result section and present it before results as a separate subsection titled: “theoretical background: master sintering curve analysis.”

3.       I would suggest presenting the results section in form of “3.3. First Principle Study on densification of Os nanocrystals”. More importantly, in there, the authors need to provide some quantified table or graphs for better summary of quantified results in the results section.

Author Response

Response to Reviewer 3 Comments

Dear reviewers:

Thank you for your comments on our manuscript entitled " Densification Mechanism of Os: Master Sintering Curve Fitting and Frist Principle Calculation." Those comments are constructive for revising and improving our paper, as well as the essential guiding significance to other research. We have studied the comments carefully and made corrections which we hope meet with approval. The primary corrections are in the manuscript, and the responses to your comments are as follows.

Point 1. I would suggest using the full name of the Osmium element throughout the manuscript instead of abbreviation “Os”. This include the title and abstract of the manuscript.

Response:

Thank you very much for your suggestion. The relevant content has been modified according to your suggestion.

Point 2. I would suggest separating the theoretical background, i.e., 3.2. MSC of Nano-sized Osmium, from the result section and present it before results as a separate subsection titled: “theoretical background: master sintering curve analysis.”

Response:

Thank you very much for your suggestion. The relevant content has been modified according to your suggestion. We have re-divided the sections of the full text and made some minor revisions.

Point 3. I would suggest presenting the results section in form of “3.3. First Principle Study on densification of Os nanocrystals”. More importantly, in there, the authors need to provide some quantified table or graphs for better summary of quantified results in the results section.

Response:

Thank you very much for your suggestion. The relevant content has been modified according to your suggestion. And Table.3, as a summary of quantified results, has been added to the manuscript.

Once again, thank you very much for your comments and suggestions.

Round 2

Reviewer 1 Report

I would like to thank the authors for the corrections which improved the quality of the paper significantly. I believe the paper can be accepted for publications.  Also, I like the comparison made by the authors between predicted densities and experimental results in Table 1. 

Please check the typesetting of the paper ( Eq 5. And the photo above it) and Table 2. Also, reporting "n" is enough since 1/n is the reciprocal values and reporting it seems redundant to me.

Author Response

Response to Reviewer 1 Comments

Dear reviewers:

Thank you for your comments on our manuscript entitled "Study on the Densification of Osmium by Experiment and First Principle Calculations."

We have studied the comments and made corrections following your guidance, and the modifications are marked in red in the last version. At the same time, the typesetting format has also been revised according to the template of the journal. 

Finally, thank you again for your valuable comments on improving our paper.

Best wishes!

Reviewer 2 Report

I recommend to the authors checking the edition after to accept all changes.

The quality of the manuscript has been improved. It can be published.

Author Response

Response to Reviewer 2 Comments

Dear reviewer:

Thank you for your comments on our manuscript entitled "Study on the Densification of Osmium by Experiment and First Principle Calculations."

We have checked the article and made appropriate corrections based on your suggestions. Meanwhile, the typesetting format has also been revised according to the template of the journal.

Finally, thank you again for your valuable comments on improving our paper.

Best wishes!

Reviewer 3 Report

The previously provided comments were addressed satisfactorily.

Author Response

Response to Reviewer 2 Comments

Dear reviewer:

Thank you again for your comments on our manuscript entitled "Study on the Densification of Osmium by Experiment and First Principle Calculations."

We have checked the article and made appropriate corrections. Meanwhile, the typesetting format has also been revised according to the template of the journal.

Finally, thank you again for your valuable comments on improving our paper.

Best wishes!
